# Association Between Movement Behaviors and Health-Related Quality of Life in Adolescents: A Cross-Sectional Study

**DOI:** 10.3390/ijerph22060969

**Published:** 2025-06-19

**Authors:** Pedro Henrique Garcia Dias, Maria Carolina Juvêncio Franscisquini, Thais Maria de Souza Silva, Géssika Castilho dos Santos, Rodrigo de Oliveira Barbosa, Jadson Márcio da Silva, Antonio Stabelini Neto

**Affiliations:** Postgraduate Program in Human Movement Sciences, Universidade Estadual do Norte do Paraná (UENP), Jacarezinho 86400-000, Paraná, Brazil; ph.garciadias@gmail.com (P.H.G.D.); maria.carolina@hotmail.com (M.C.J.F.); thais.msouza@outlook.com (T.M.d.S.S.); gessika.castilho@gmail.com (G.C.d.S.); rodrigo.oli.barbos@gmail.com (R.d.O.B.); jadson_marcio@hotmail.com (J.M.d.S.)

**Keywords:** physical activity, sedentary behavior, sleep, quality of life, young people

## Abstract

Health-related quality of life (HRQoL) is a multidimensional construct that encompasses physical, emotional, psychological, and social domains, according to an individual’s perception. Studies have indicated that lifestyle-related factors, such as engaging in physical activity (PA), reducing screen time (ST), and maintaining adequate sleep duration, may contribute to improved HRQoL in adolescents. The present study aimed to examine the associations of the 24 h movement behaviors (PA, ST, and sleep duration) with physical and psychological well-being, and HRQoL in adolescents. This study included 746 adolescents of both sexes, aged 11 to 15 years, enrolled in public schools. Sleep duration was assessed through a specific question related to habitual bedtime and wake-up time. ST was evaluated using a question regarding the daily time spent using recreational electronic devices. PA was measured using accelerometers (ActiGraph GT3X-BT). The KIDSCREEN questionnaire was used to assess physical and psychological well-being and HRQoL. Generalized linear models were used for statistical analysis. Significant associations were observed between meeting sleep duration recommendations and higher HRQoL scores (β = 1.05, 95% CI: 1.01–1.08), as well as psychological well-being (β = 1.07, 95% CI: 1.03–1.11). Additionally, adherence to ST recommendations was significantly associated with higher physical well-being scores (β = 1.07, 95% CI: 1.01–1.14). Concerning the combination of adherence to guidelines, meeting both sleep duration and ST recommendations was significantly associated with higher HRQoL scores (β = 1.08, 95% CI: 1.02–1.15), physical well-being (β = 1.11, 95% CI: 1.03–1.21), and psychological well-being (β = 1.09, 95% CI: 1.01–1.18). The findings of this study highlight the importance of adhering to the 24 h movement guidelines, which may contribute to improved adolescent well-being.

## 1. Introduction

Quality of life (QoL) is defined as an individual’s perception of their position in life within the context of the culture and value system they live in, and in relation to their goals, expectations, standards, and concerns [1]. QoL is a broad concept that complexly integrates physical health, psychological state, level of independence, social relationships, personal beliefs and their relationships with the defining characteristics of the environment [1]. In turn, health-related quality of life (HRQoL) is understood as a subset of QoL, defined as a multidimensional construct encompassing the physical, emotional, psychological, and social domains, according to the individual’s own perception [2]. HRQoL is a widely used indicator to assess the impact of health on daily life, especially in pediatric and adolescent populations [3]. The primary distinction between these two concepts is that, while QoL has a broader approach and encompasses an individual’s overall well-being, HRQoL focuses on aspects of life directly influenced by health conditions, such as the physical, psychological, social, family, and school aspects of well-being and functional capacity [2,4].

HRQoL in adolescence can be influenced by a range of lifestyle factors, including physical activity (PA), sedentary behavior, and sleep [5]. Currently, these three lifestyle behaviors are recognized in the literature as movement behaviors, which, when combined, reflect the 24 h daily cycle [6]. While these behaviors operate separately, their significance for the physical and mental well-being of youth is acknowledged in existing research [7].

Moreover, it is important to consider that these behaviors share the limited time available in a day, so an increase in one necessarily implies a reduction in the time allocated to the others (i.e., more time spent in sedentary behaviors may mean less time available for physical activity or sleep) [6]. Therefore, the interdependence between these movement behaviors throughout the day should be taken into account when investigating their effects on adolescent health and quality of life.

Regular PA has been associated with several health benefits in adolescents, such as improved cardiorespiratory fitness [8], body composition [9], metabolic profile [10] and mental health [11]. In contrast, studies have shown that excessive screen time (ST) is associated with poor body composition [12], increased risk factors for cardiometabolic conditions, negative behavioral traits [13], lower physical fitness [14] and worse mental health [15]. Regarding sleep, studies have shown that shorter sleep duration is associated with unfavorable indicators of adiposity [16], poor emotional regulation (stress, anxiety, depression) [17], lower academic performance [18], and behavioral problems (drugs or alcohol use, violence and scholar absenteeism) [19].

Within this context, research indicates that HRQoL may be linked to these movement behaviors. Longitudinal studies have shown that higher levels of PA are associated with better HRQoL in physical, mental, and social dimensions [20]. Regarding ST, a systematic review showed a significant association between higher ST and poorer HRQoL [21]. Moreover, longitudinal studies have revealed consistent evidence that increased ST correlates with lower scores in various HRQoL domains, including physical, psychosocial, mental, and emotional [22]. Regarding sleep, studies have identified a significant association between meeting sleep recommendations and higher HRQoL [23]. Additionally, regular compliance with sleep recommendations on weekdays and weekends was associated with improved HRQoL in Norwegian adolescents [24]. Otherwise, persistent insufficient sleep during the weekdays and weekends was associated with lower HRQoL [25].

Although studies analyze movement behaviors and their relationship with HRQoL in isolation, it is important to notice that individuals engage in a set of behaviors simultaneously, and this interaction can influence the effect of each behavior. Findings of studies suggest that engaging in integrated movement behaviors is associated with better physical health indicators, such as improved cardiorespiratory and muscular fitness [8] and cardiometabolic health [26]. Cross-sectional studies also reported an association between engaging in two or more behaviors and a lower risk of anxiety and depression in adolescents [11]. Additionally, favorable associations were found with mental health indicators, such as impulsivity, psychological well-being, and pro-social behavior [27].

Although international literature has advanced in the knowledge of the associations between the 24 h movement behaviors and adolescents’ health, few studies have analyzed the combined relationship of these behaviors in Brazilian adolescents. Of these studies, adherence to the 24 h movement behaviors guidelines was associated with mental health [28,29] and cardiometabolic markers [30]. Therefore, although these studies represent progress by adopting an integrated approach aligned with international guidelines, there is still a gap in understanding the combined effects on quality of life and well-being of Brazilian adolescents.

In light of the aforementioned points, it is evident that the literature still lacks studies that analyze movement behaviors and HRQoL in an integrated approach. Furthermore, most studies investigating this relationship in adolescents come from high-income countries [31]. To our knowledge, only two studies have been conducted on this topic among Brazilian adolescents [23,32]. Therefore, further research exploring data from the South American population is needed, as this study will not only contribute to the body of evidence on the topic but also provide information for the development of policies aimed at promoting adolescent health. Thus, this study aimed to verify the association of the 24 h movement behaviors (PA, ST, and sleep duration) with indicators of physical and psychological well-being, and HRQoL in adolescents. It was hypothesized that meeting the 24 h movement behavior guidelines would be associated with improved quality of life in these adolescents.

## 2. Materials and Methods

### 2.1. Study Design and Ethical Procedures

This study adopted a school-based cross-sectional design, following Strengthening the Reporting of Observational Studies in Epidemiology (STROBE) recommendations [33]. The study procedures were approved by the Research Ethics Committee of the State University of Northern Paraná, Brazil (Registration No. 6.566.645).

### 2.2. Population and Sample

Jacarezinho is a city located in the northern region of the state of Paraná, with a medium Human Development Index (HDI = 0.743). According to the 2022 census conducted by the Brazilian Institute of Geography and Statistics (IBGE), the city of Jacarezinho has 40,375 inhabitants. Students aged 11 to 15 years of both sexes (7th, 8th, and 9th grades) were invited to participate. The inclusion criteria for taking part in the study were (I) written consent to participate from both parents/guardians and students; (II) being regularly enrolled in the 7th, 8th, or 9th grade. II. Exclusion criteria for the study included (I) having any type of physical or mental disability that could interfere with understanding the questionnaires; (II) failure to properly fill in the questionnaire information or provide valid accelerometer data for analysis (for those who used the accelerometer).

### 2.3. Procedures, Instruments, and Data Collection Techniques

All the assessments were conducted in the school setting by trained researchers during physical education classes. Data collection was carried out starting in August of 2023 and ending in June of 2024. The data collection proceeded as follows: (1) questionnaire with personal information, recreational screen time, total daily sleep time, and health-related quality of life; (2) after completing the questionnaires, students went to a private room for anthropometric assessments; (3) distribution and guidance about the use of the accelerometers.

### 2.4. Physical Activity

Moderate-to-vigorous physical activity (MVPA) was measured using the ActiGraph GT3X^+^ (Pensacola, FL, USA) triaxial accelerometer. The accelerometers were set to collect data at a sampling rate of 80 Hz. It is important to note that accelerometer use was assigned to a random selection of 50% of the total sample, due to the limited number of devices available. The distribution process was as follows: classroom selection by random draw, device programming, and delivery of devices. The adolescents were instructed to wear the accelerometer on the hip, positioned at the level of the anterior superior iliac spine, for seven consecutive days, removing it only to sleep, bath or partake in water-related activities. After the devices were returned, the data were transferred to a computer and stored. Data were processed using ActLife software version 6.13. Inactivity periods were defined as ≥30 consecutive minutes of “0” counts. Only participants with valid recordings (i.e., >8 h per day for at least 3 days, including two weekdays and one weekend day) were considered in the analyses. The average daily time (in minutes) spent in MVPA was estimated using the cut-points proposed by Evenson [34]. To assess adherence to the 24 h movement guidelines, engagement in ≥60 min of MVPA per day was adopted [6].

### 2.5. Screen Time

This outcome was measured through a subjective self-reported method, by the question *“On a typical weekday, how much time do you spend watching TV, playing video games, or using a computer or cellphone during your leisure time?”* Responses were structured into nine options, ranging from “I don’t spend time watching TV, playing video games, or using a computer or cellphone” to “More than 7 h a day.” It is important to highlight that this assessment method is widely used in population-based studies [23]. For recreational ST, a cut-off point of no more than two hours per day was adopted [6].

### 2.6. Sleep

Sleep duration was assessed through the following questions: “*What time do you usually go to sleep?*” and “*What time do you usually wake up?*” Based on these responses, sleep time was calculated as the difference between bedtime and wake-up time. Adequate sleep duration was defined as 9 to 11 h per night for ages 5 to 13, and 8 to 10 h per night for adolescents aged 14 to 17 [6].

### 2.7. Health-Related Quality of Life

QoL was assessed using the KIDSCREEN-27 questionnaire, which is validated for Brazilian adolescents [35]. The questionnaire measures five dimensions: (1) physical well-being, (2) psychological well-being, (3) autonomy and parent relation, (4) peers and social support, and (5) school environments. For this study, the physical and psychological well-being dimensions were used. The physical well-being dimension presents an intraclass correlation coefficient (ICC) of 0.75, reproducibility of 76.4 and internal consistency of 77.7. The psychological well-being dimension presents an ICC of 0.81, reproducibility of 83.5 and internal consistency of 85.7. HRQoL was assessed using the KIDSCREEN-10 [36] questionnaire. Its psychometric properties in adolescents aged 12 to 18 years are considered adequate, with a Cronbach’s alpha of 0.81 and a test–retest ICC of 0.69. Responses are coded into a score ranging from 10 to 50, with the highest scores representing better HRQoL.

### 2.8. Anthropometric Variables

Measurements of body weight (kg) and body height (cm) followed a standardized process and were performed by qualified staff. Height was measured using a stadiometer (Lider; model LD1015, São Paulo, Brazil) with a resolution of 0.1 cm. Body weight was measured using a digital scale (Tanita; model BC-558, Tokyo, Japan) with a resolution of 0.1 kg. Body Mass Index (BMI) was calculated using the formula BMI = body mass (kg)/height (m^2^) [37].

### 2.9. Data Analysis

Participants’ general characteristics were described as means and standard deviations for numerical variables, and frequency distributions for categorical variables. To test the hypothesis that adherence to the 24 h movement behavior guidelines is associated with improved quality of life in adolescents, the participants were classified into the following categories: do not meet any recommendations; meets only one of the three recommendations (PA; ST; or sleep); meets combinations of two recommendations (PA + ST; PA + sleep; or ST + sleep); or meets all three recommendations. To analyze the associations between the movement behaviors and scores of physical well-being, psychological well-being, and HRQoL, generalized linear models were used. Combined analyses were performed using the group that did not meet any of the recommendations as a reference. Analyses were adjusted by sex and age. Data were analyzed using the Statistical Package for the Social Sciences (SPSS), version 25.0, with a significance level of *p* < 0.05.

## 3. Results

The present study recruited 815 adolescents, and after applying the exclusion criteria, the final sample consisted of 746 participants (51.7% female; age 13.43 ± 1.10). Table 1 presents the characteristics of the participants. Overall, when comparing the accelerometer subsample with the total sample, statistically significant differences were observed for age (*p* < 0.00), height (*p* < 0.02), and sleep duration (*p* < 0.00). Analyzing the total sample, it was found that 49.8% did not meet any of the recommendations, while 11.9% met two recommendations, and 2.3% of adolescents met all three recommendations.

Table 2 presents the associations between individual and combined movement behaviors with HRQoL, physical well-being, and psychological well-being. The individual analyses revealed significant associations between meeting the sleep duration recommendation and higher scores for HRQoL (β = 1.05, 95% CI: 1.02–1.08) and psychological well-being (β = 1.07, 95% CI: 1.03–1.11). Additionally, meeting the PA recommendation was significantly associated with higher scores of physical well-being (β = 1.14, 95% CI: 1.06–1.22) and psychological well-being (β = 1.08, 95% CI: 1.01–1.15). After adjusting for sex and age, the analyses still showed a significant association between meeting the sleep duration recommendation and higher scores for HRQoL (β = 1.05, 95% CI: 1.01–1.08) and psychological well-being (β = 1.07, 95% CI: 1.03–1.11). Moreover, meeting the ST recommendation was significantly associated with higher physical well-being scores (β = 1.07, 95% CI: 1.01–1.14).

When examining combinations of adherence to 24 h movement behaviors guidelines, meeting both the PA + ST recommendations was associated with higher physical well-being scores (β = 1.16, 95% CI: 1.00–1.35). Meeting the sleep duration + ST recommendations was significantly associated with higher scores in HRQoL (β = 1.08, 95% CI: 1.02–1.15), physical well-being (β = 1.10, 95% CI: 1.02–1.19), and psychological well-being (β = 1.09, 95% CI: 1.01–1.18). After adjusting for sex and age, meeting the sleep duration + ST recommendations remained significantly associated with higher scores in HRQoL (β = 1.08, 95% CI: 1.02–1.15), physical well-being (β = 1.11, 95% CI: 1.03–1.21), and psychological well-being (β = 1.09, 95% CI: 1.01–1.18).

## 4. Discussion

The present study aimed to examine the associations between meeting the 24 h movement behavior guidelines (PA, ST, and sleep duration) and physical well-being, psychological well-being, and HRQoL in school-aged adolescents. The main findings revealed that meeting the 24 h movement behaviors guidelines, particularly ST and sleep duration, were associated with higher perceptions of HRQoL, physical well-being, and psychological well-being.

When analyzing the associations of movement behaviors individually, MVPA was associated with both physical and psychological well-being. However, these associations lost significance after adjusting for sex and age. Our findings align with previous studies that also found no significant association between PA—measured either objectively or subjectively—and HRQoL in adolescents [23,38]. Nevertheless, prior research has demonstrated significant associations between regular PA and HRQoL [39]. Different mechanisms may explain this relationship. On a neurobiological level, PA causes structural and functional brain changes resulting in increased endorphin release and, consequently, improved psychological well-being. On a psychosocial level, participation in physical activities creates opportunities for interaction and social acceptance, which positively influence factors such as body image, independence, and perceived competence—key contributors to well-being. Therefore, although the literature has shown that higher levels of PA are associated with a lower risk of depressive symptoms [40], improved cardiorespiratory fitness [8], improved metabolic profiles [30], and improved subjective well-being [41], our results were not able to confirm the beneficial influence of PA on HRQoL in adolescents.

Regarding ST, our results indicated that adolescents with less than two hours of recreational ST per day had a significant association with higher scores of physical well-being. In line with our findings, previous studies have also reported significant associations between lower ST and better physical health [42]. Furthermore, studies have found inverse associations between higher ST and both lower psychological well-being [43] and HRQoL [23]. Part of these results may be explained by the substitution of time spent on health activities, such as PA, with sedentary behaviors (time-displacement theory) [44], as well as the passive and often solitary nature of screen-based activities, which can limit or replace social interaction with friends, family, and parents [45], negatively affecting psychological well-being and HRQoL [46]. Moreover, studies have shown that adolescents with higher ST may exhibit aggressive behaviors, learning difficulties, attention-deficit/hyperactivity disorder, and sleep disturbances [45,47].

The results of the current study also showed a significant association between compliance with the sleep recommendation and better perception of HRQoL and psychological well-being. Our findings are consistent with previous research demonstrating the isolated association between sleep duration and QoL, reinforcing the benefits of sleep for adolescents’ physical, psychological, and social health [23,48]. Part of our results can be explained by the fact that social activities and other behavioral patterns often lead adolescents to adopt nighttime habits, while school schedules require them to wake up early and be fully alert in the morning. This dynamic results in a reduced sleep duration during the week, leading to a persistent sleep debt. In this context, the reduction in night sleep, caused by shortened or irregular sleep schedules, has been associated with excessive sleepiness, aggressive behavior, and impaired cognitive performance [49,50]. In addition, sleeping less than recommended can negatively affect mood, cognitive processing, memory consolidation, QoL, social interactions, and both externalizing and internalizing problems [49,51].

Considering the combined analyses, the present study found that adolescents who met both ST and sleep duration recommendations showed better perceptions of physical well-being, psychological well-being, and HRQoL. In line with our findings, some studies showed that meeting two or more lifestyle recommendations results in higher scores for physical well-being, psychological well-being, and HRQoL compared to adolescents who met none [23,39]. One explanation for the significant association of this combination (ST + sleep) is that screen use, especially close to bedtime, can directly disturb sleep, causing physiological arousal that makes it difficult for children and adolescents to relax. As a result, individuals who do not follow these recommendations and spend more time on screens tend to sleep less, take longer to fall asleep, go to bed later and experience changes in sleep architecture, which compromises sleep quality [10]. Because these factors are interdependent, they further hinder adherence to healthy sleep behaviors. These negative effects generate a cascading impact, in which non-adherence to these behaviors can contribute to poorer well-being and HRQoL.

Regarding the combination of PA and ST, our results did not find a significant association. One of the reasons may be that only 2.1% of adolescents met the combined PA + ST recommendations. Another plausible explanation is that adolescents spent more time in front of screens, which may have negatively impacted their participation in PA. According to the time-displacement theory, when adolescents spend more time playing video games or using computers, they tend to reduce the time available for PA [48], which negatively affects their health and well-being [52]. In the same way, we did not find a significant association for PA + sleep combination with well-being and HRQoL. Adolescents who start sleeping later tend to practice less PA the next day, which favors an increase in sedentary behavior and further damages the relationship between PA and sleep [53]. This negative cycle can have a significant impact on HRQoL, as insufficient PA and sleep affect both physical and psychological well-being [54].

The findings of the present study are consistent with those of Guedes et al. [23], who identified positive associations between lower screen time, adequate sleep duration, and better perceptions of HRQoL in adolescents. Additionally, in both studies, PA was not significantly associated with HRQoL, which suggests that the effect of PA on quality of life is dependent on its interaction with other behaviors. Furthermore, Guedes et al. [23] emphasize the importance of simultaneous adherence to multiple recommendations—particularly adequate sleep and limited screen time—is associated with better indicators of physical well-being, psychological well-being, and HRQoL. Indeed, it is essential that public health strategies prioritize the development of integrated policies that promote adherence to these recommendations, especially with respect to reducing screen time and increasing sleep duration. This approach is particularly relevant during adolescence, a critical period for the development of health-related habits that tend to persist into adulthood and influence long-term health outcomes [55].

Finally, the results of the present study should be interpreted in light of its limitations. First, although the results support the hypothesis of a positive association between the 24 h movement behaviors guidelines and well-being and HRQoL, the cross-sectional study design does not allow causal inferences. Second, ST, sleep duration, well-being and HRQoL variables were self-reported through questionnaires, which are subject to known limitations such as recall bias. It is also important to highlight that health behavior choices during adolescence are strongly influenced by various factors, such as family and school environments [56], as well as social desirability bias. Lastly, although this study used accelerometry to measure PA, the sample size of adolescents with valid data was limited, which may have affected the statistical power of analyses involving this behavior.

Nonetheless, the study has notable strengths. It used a representative sample of adolescents and employed objective measurement of PA using accelerometry. Moreover, this study highlights the relevance of the 24 h movement behaviors guidelines for health prevention policies and contributes to expanding the body of evidence on their relationship with well-being and HRQoL in Brazilian adolescents.

## 5. Conclusions

The findings of this study highlight that meeting the recommendations of the 24 h movement behavior guidelines, particularly ST and sleep duration, is positively associated with adolescents’ perceived well-being and HRQoL. Moreover, the combined effect of ST and sleep duration had an additive influence on physical well-being, psychological well-being, and HRQoL. These findings are significant, given the crucial role of maintaining adequate HRQoL throughout life. Therefore, there is a clear need for longitudinal studies and educational interventions, as well as public health strategies and prevention programs specifically targeting adolescents.

## Figures and Tables

**Table 1 ijerph-22-00969-t001:** Sample characteristics.

Variables	Total Sample (n = 746)	Accelerometer Sample (n = 349)
**Age (years)**	13.4 ± 1.1	13.2 ± 1.0 *
**Height (cm)**	158.2 ± 10.4	157.2 ± 9.6 *
**Body Mass Index (kg/m^2^)**	21.1 ± 4.8	21.0 ± 4.8
**Sex**		
Male, n (%)	384 (48.3)	151 (44.4)
Female, n (%)	411 (51.7)	189 (55.6)
**Screen Time** (Hours/day)	5.5 ± 2.3	5.5 ± 2.2
**Sleep** (Hours/night)	8.1 ± 1.6	7.8 ± 1.5 *
**MVPA** (Min/day)	-	39.8 ± 22.0
HRQoL (Score 10–50)	35.5 ± 7.2	33.0 ± 7.1
**Physical Well-Being** (Score 5–25)	16.8 ± 4.3	17.0 ± 4.2
**Psychological Well-Being** (Score 7–35)	25.0 ± 5.9	25.1 ± 5.8
**Physical Activity Recommendations**		
Yes	-	65 (18.7)
No	-	282 (81.3)
**Screen Time Recommendations, n (%)**		
Yes	83 (11.2)	32 (9.8)
No	660 (88.8)	294 (90.2)
**Sleep Recommendations, n (%)**		
Yes	235 (33)	88 (28.2)
No	447 (67)	224 (71.8)
**24 h Movement Guidelines, n (%)**		
None, n (%)	-	154 (49.8)
ST + SLP, n (%)	36 (5.1)	19 (4.7)
PA + SLP, n (%)	-	16 (5.1)
PA+ ST, n (%)	-	7 (2.1)
PA + ST + SLP, n (%)	-	7 (2.3)

* Statistical significance: *p* < 0.05; abbreviations: MVPA = moderate-to-vigorous physical activity; HRQoL = health-related quality of life; ST = screen time; SLP = sleep duration; PA = physical activity; physical activity recommendation: ≥60 min/day; screen time recommendation: ≤2 h/day; sleep recommendation: 9–11 h/night (ages 5–13); 8–10 h/night (ages 14–17).

**Table 2 ijerph-22-00969-t002:** Isolated and combined associations between meeting the 24-h movement behavior guidelines, health-related quality of life, and well-being in adolescents.

Variables	HRQoL		Physical Well-Being		Physicological Well-Being	
	Crude β (95% CI)	Adjusted β (95% CI)	Crude β (95% CI)	Adjusted β (95% CI)	Crude β (95% CI)	Adjusted β (95% CI)
Meets ST recommendations						
Yes	1.02 (0.96–1.08)	1.02 (0.96–1.09)	1.05 (0.98–1.12)	**1.07 (1.0–1.14)**	1.02 (0.95–1.09)	1.04 (0.97–1.11)
No	1	1	1	1	1	1
Meets Sleep recommendations						
Yes	**1.05 (1.02–1.08)**	**1.05 (1.01–1.08)**	1.01 (0.97–1.05)	1.01 (0.97–1.05)	**1.07 (1.03–1.11)**	**1.07 (1.03–1.11)**
No	1	1	1	1	1	1
Meets PA recommendations						
Yes	1.05 (0.99–1.11)	1.0 (0.94–1.06)	**1.14 (1.06–1.22)**	1.07 (0.99–1.15)	**1.08 (1.01–1.15)**	1.0 (0.94–1.07)
No	1	1	1	1	1	1
Meets PA + ST recommendations						
Yes	1.06 (0.94–1.20)	1.01 (0.87–1.17)	**1.16 (1.0–1.35)**	1.08 (0.92–1.28)	1.03 (0.88–1.22)	0.95 (0.81–1.12)
No	1	1	1	1	1	1
Meets PA + sleep recommendations						
Yes	1.04 (0.94–1.16)	1.01 (0.92–1.11)	1.08 (0.96–1.22)	1.03 (0.93–1.14)	1.02 (0.90–1.16)	0.97 (0.88–1.08)
No	1	1	1	1	1	1
Meets ST + sleep recommendations						
Yes	**1.08 (1.02–1.15)**	**1.08 (1.02–1.15)**	**1.10** **(1.02–1.19)**	**1.11 (1.03–1.21)**	**1.09 (1.01–1.18)**	**1.09 (1.01–1.18)**
No	1	1	1	1	1	1
Meets all three recommendations						
Yes	0.95 (0.81–1.11)	0.95 (0.80–1.13)	1.03 (0.84–1.27)	1.02 (0.85–1.23)	0.94 (0.79–1.12)	0.95 (0.80–1.12)
No	1	1	1	1	1	1

Abbreviations: HRQoL = health-related quality of life; ST = screen time; PA = physical activity; 95% CI: 95% confidence intervals; physical activity recommendation: ≥60 min/day; screen time recommendation: ≤2 h/day; sleep recommendation: 9–11 h/night (ages 5–13); 8–10 h/night (ages 14–17). 1—reference group (does not meet the recommendation); crude analyses: generalized linear models without adjustments by confusion variables; adjusted analyses: generalized linear models adjusted by sex and age; statistical significance: highlighted in bold at *p* < 0.05.

## Data Availability

The dataset of the present study is available from the corresponding author upon reasonable request.

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
