# Peer review of "Association Between Movement Behaviors and Health-Related Quality of Life in Adolescents: A Cross-Sectional Study"

_ijerph, 2025, doi:10.3390/ijerph22060969_

Round 1

Reviewer 1 Report

Comments and Suggestions for Authors

Thank you for the opportunity to review "Association Between Movement Behaviors and Health-Related Quality of Life in Adolescents: A Cross Sectional Study" which examine self-report physical and psychological wellbeing, as well as screen time use, sleep duration and physical activity in 745 adolescents. The study is interesting and of value. I have made some suggestions for improvements below.

The introduction is largely relevant, well justified and provides sound rationale. It is important in this area to consider that these particular behaviors all come at a time cost to others. For example, if we spend more time sleeping, we will inevitably spend less time physically active or sedentary. Even if the goal of your study is not to explore this, I feel this is an important piece of the dynamic that you must express.

Could you please justify why, given you used accelerometer in your study, participants were asked to remove it for sleep? As the participants already had the devices, had you also recorded sleep you would have collect 24 hours movement data which would have significantly enhanced this analysis. This decision must be justified in the manuscript.

For the screen time measure, have you considered whether these activities were sedentary or not? As your goal was to understand movement behaviors, these screen time activities may have actually been active (i.e., video games with fitness component). Additionally, your question does not account for time spent using screen for educational purposes. The guidelines around screen time use explicitly state recreational screen time use. It is likely your participants may have included the time they spent on screens during school hours also.

Did you sleep duration questionnaire consider the differences between weekday and weekend bedtimes? Evidence highlights that bedtimes vary significantly between weeknights and weekdays in this age group (https://doi.org/10.1016/j.sleh.2021.07.008)

For your data analysis it would be beneficial if you could justify why you used the “meeting guidelines” (especially dichotomising variables) rather than continuous variables of sleep duration etc. Given you have such small samples meeting guidelines, thus impacting the generalisability of your results. These decisions need to be well justified in the manuscript.

In your discussion you wrote “The main findings revealed that movement behaviors, particularly adequate ST and sleep duration were associated with higher perceptions of HRQoL, physical well-being, and psychological well-being”. I am unclear what you mean by adequate screen time? I also do not feel that you finding “In this context, our study adds to existing literature by suggesting that the combination of multiple health-related behaviors has a greater impact on HRQoL in adolescents.” as your results show a combination of the three behaviors is not significant. Additionally, as you have not presented effect sizes you cannot draw this conclusion.

Throughout the manuscript and in particular the tables there are typos (“Phycological”), errors in spelling, words that are not in English (“Sim”, Não”). Additionally, Table 2 is poorly formatted and it is difficult to see which results are for which analysis. Please edit the Table 2 header/footnote to be clearer on the analysis presented, are these the models with covariates?

Comments on the Quality of English Language

Please make sure all words are in English.

Author Response

Reviewer 1

  1. The introduction is largely relevant, well justified and provides sound rationale. It is important in this area to consider that these particular behaviors all come at a time cost to others. For example, if we spend more time sleeping, we will inevitably spend less time physically active or sedentary. Even if the goal of your study is not to explore this, I feel this is an important piece of the dynamic that you must express.

Response: Thank you for your comment. A sentence was added to clarify the interdependence among the 24-hour movement behaviors (P.2, Lines: 74-81).

  1. Could you please justify why, given you used accelerometer in your study, participants were asked to remove it for sleep? As the participants already had the devices, had you also recorded sleep you would have collect 24 hours movement data which would have significantly enhanced this analysis. This decision must be justified in the manuscript.

Response: We appreciate your point out. We used the Canadian 24h movement behavior guidelines, and the recommendation about recreational screen time is <2 hours per day, so the accelerometer cannot provide this information. Moreover, we configured the accelerometer to be to worn on the hip, because it´s more accurate at capturing whole-body movement, therefore, the adolescents should remove it for sleep. In studies that used the accelerometer to measure sleep, it was worn on the wrist.

  1. For the screen time measure, have you considered whether these activities were sedentary or not? As your goal was to understand movement behaviors, these screen time activities may have actually been active (i.e., video games with fitness component). Additionally, your question does not account for time spent using screen for educational purposes. The guidelines around screen time use explicitly state recreational screen time use. It is likely your participants may have included the time they spent on screens during school hours also.

Response: Thank you for the comment. There really was a typo in the text. In fact, the adolescents were asked about the time spent in screen time exclusively during leisure time. The correct question was introduced.

On a typical weekday, how much time do you spend watching TV, playing video games, or using a computer or cellphone during your leisure time?” (P.6, Lines: 212-214).

  1. Did you sleep duration questionnaire consider the differences between weekday and weekend bedtimes? Evidence highlights that bedtimes vary significantly between weeknights and weekdays in this age group (https://doi.org/10.1016/j.sleh.2021.07.008)

Response: Thank you for your comment. The authors agree that this may be a limitation of the present study, since we just asked about the sleep duration during the week. However, although some studies have shown differences in the sleep duration between weekdays and weekends in adolescents, there were no signicant effects on the outcome variables. Additionally, when the researchers assess the sleep duration separately by week or weekend, they need to adjust it by the number of days (week 5 days – higher weight / weekend 2 days – lower weight), so we believe that assessing just the week is representative of the adolescents´ sleep behavior and had no effect on the analyses.

  1. For your data analysis it would be beneficial if you could justify why you used the “meeting guidelines” (especially dichotomizing variables) rather than continuous variables of sleep duration etc. Given you have such small samples meeting guidelines, thus impacting the generalizability of your results. These decisions need to be well justified in the manuscript.

Response: We appreciate your suggestion. We agree with your comment that we could have analyzed the data using statistical approaches for continuous variables, however, we intended to test the hypothesis that adherence to the Canadian 24-hour movement behavior guidelines is associated with improved quality of life in adolescents. This rationale has been included in the text.   (P.4, Lines: 145-147; P.7, Lines: 260-263).

  1. In your discussion you wrote “The main findings revealed that movement behaviors, particularly adequate ST and sleep duration were associated with higher perceptions of HRQoL, physical well-being, and psychological well-being”. I am unclear what you mean by adequate screen time? I also do not feel that you finding “In this context, our study adds to existing literature by suggesting that the combination of multiple health-related behaviors has a greater impact on HRQoL in adolescents.” as your results show a combination of the three behaviors is not significant. Additionally, as you have not presented effect sizes you cannot draw this conclusion.

Response: Thank you for your comment. The sentence was rewritten to make these statements clearer and in line with our findings.   (P.12, Lines: 358-361).

  1. Throughout the manuscript and in particular the tables there are typos (“Phycological”), errors in spelling, words that are not in English (“Sim”, Não”). Additionally, Table 2 is poorly formatted and it is difficult to see which results are for which analysis. Please edit the Table 2 header/footnote to be clearer on the analysis presented, are these the models with covariates?

Response: Thank you for your suggestions. A complete English review was done throughout the manuscript, as well as the tables were edited for greater clarity and better understanding.

Reviewer 2 Report

Comments and Suggestions for Authors

The present study aimed to examine the associations between 24-hour movement behaviors (physical activity, sedentary time, and sleep duration) and physical well-being, psychological well-being, and health-related quality of life (HRQoL) in adolescents. The authors used a large, representative sample, and the manuscript is generally well written. However, there are some areas that require improvement or further clarification.

Introduction:

Please consider providing a more comprehensive overview of previous Brazilian studies that have examined 24-hour movement behaviors in adolescents.

Highlight the specific research gaps in the Brazilian context to justify the relevance of the current study.

Methods:

Consider enhancing the statistical analysis by applying compositional data analysis (CoDA), which is more appropriate for examining 24-hour movement behaviors, given their co-dependent.

Results:

Line 234: Could the statistically significant differences observed be partially explained by issues related to the randomization or sampling process? Please clarify or discuss this possibility.

Discussion:

It would strengthen your discussion to compare your findings with those of Guedes et al. (2023).

I recommend focusing your discussion more clearly on the associations between adherence to the 24-hour movement behavior guidelines (physical activity, sedentary behavior, and sleep duration) and physical/psychological well-being and HRQoL, as this aligns directly with your study objectives.

You could also benefit from comparing your findings with both Brazilian and European studies, highlighting similarities and differences in behavioral patterns and their associations with well-being outcomes.

Table 2: Please include a clear and informative legend below the table.

Author Response

Reviewer 2

  1. Please consider providing a more comprehensive overview of previous Brazilian studies that have examined 24-hour movement behaviors in adolescents. Highlight the specific research gaps in the Brazilian context to justify the relevance of the current study.

Response: Thank you for your comments. More comprehensive information regarding Brazilian studies on 24-hour movement behaviors was included to better contextualize the national scientific production and to reinforce the originality and relevance of the present research. (Pages 3-4; Lines: 121-131).

  1. Methods: Consider enhancing the statistical analysis by applying compositional data analysis (CoDA), which is more appropriate for examining 24-hour movement behaviors, given their co-dependent.

Response: We greatly appreciate your suggestion. We agree with your comment that compositional data analysis is more appropriate for examining 24-hour movement behaviors, however this approach works best when measuring the entire 24-hour period with the same instrument, which was not possible in current study. Furthermore, we intended to test the hypothesis that adherence to the Canadian 24-hour movement behavior guidelines is associated with improved quality of life in adolescents, which is the rationale for using this methodological approach.

Results:

  1. Line 234: Could the statistically significant differences observed be partially explained by issues related to the randomization or sampling process? Please clarify or discuss this possibility.

Response: Thank you for your comment. We believe not, since accelerometer use was determined by random sampling. It is likely that older adolescents (based on slightly lower age and height in the group with valid accelerometer data) used the devices less. Furthermore, although there were statistical differences between the groups, these differences were not impacted in the individual and combined analyses of ST and SLP, since these analyses were conducted with the total sample.

  1. Discussion:

It would strengthen your discussion to compare your findings with those of Guedes et al. (2023).

I recommend focusing your discussion more clearly on the associations between adherence to the 24-hour movement behavior guidelines (physical activity, sedentary behavior, and sleep duration) and physical/psychological well-being and HRQoL, as this aligns directly with your study objectives.

You could also benefit from comparing your findings with both Brazilian and European studies, highlighting similarities and differences in behavioral patterns and their associations with well-being outcomes.

Response:  Thank you for your suggestion. We included the main results of the study of Guedes et al. (2023) in the discussion was well as we compared our findings with the current literature. Furthermore, changes were made in order to align the interpretation of the findings with the study’s objectives.  (P.15; Lines: 462-480).

  1. Table 2: Please include a clear and informative legend below the table.

Response: Thank you for your suggestion. An informative legend was included below the table 2.

Round 2

Reviewer 1 Report

Comments and Suggestions for Authors

Thank you for addressing my previous comments.

Reviewer 2 Report

Comments and Suggestions for Authors

Dear authors,

I am satified with changes. Thank you.